

# Worsening urban ozone pollution in China from 2013 to 2017 – Part 1: The complex and varying roles of meteorology

Yiming Liu[1], Tao Wang[1]

[1]Department of Civil and Environmental Engineering, The Hong Kong Polytechnic University, Hong Kong, 999077, China

*Correspondence to*: Tao Wang (cetwang@polyu.edu.hk)

**Abstract.** China has suffered from increasing levels of ozone pollution in urban areas despite the implementation of various stringent emission reduction measures since 2013. In this study, we conducted numerical experiments with an up-to-date regional chemical transport model to assess the contribution of the changes in meteorological conditions and anthropogenic emissions to the summer ozone level from 2013 to 2017 in various regions of China. The model can faithfully reproduce the

observed meteorological parameters and air pollutant concentrations and capture the increasing trend in the surface maximum daily 8-hour average (MDA8) ozone ($O_3$) from 2013 to 2017. The emission control measures implemented by the government induced a decrease in MDA8 $O_3$ levels in rural areas but an increase in urban areas. The meteorological influence on the ozone trend varied by region and by year and could be comparable to or even more significant than the impact of changes in anthropogenic emissions. Meteorological conditions can modulate the ozone concentration via direct (e.g., increasing reaction

rates at higher temperatures) and indirect (e.g., increasing biogenic emissions at higher temperatures) effects. As an essential source of volatile organic compounds that contributes to ozone formation, the variation in biogenic emissions during summer varied across regions and was mainly affected by temperature. China's midlatitude areas (25°N to 40°N) experienced a significant decrease in MDA8 $O_3$ due to a decline in biogenic emissions, especially for the Yangtze River Delta and Sichuan Basin regions in 2014 and 2015. In contrast, in northern (north of 40°N) and southern (south of 25°N) China, higher

temperatures after 2013 led to an increase in MDA8 $O_3$ concentrations via an increase in biogenic emissions. We also assessed the individual effects of changes in temperature, specific humidity, wind field, planetary boundary layer height, clouds, and precipitation on ozone levels from 2013 to 2017. The results show that the wind field change made a significant contribution to the increase in surface ozone over China by transporting the ozone downward from the upper troposphere and the lower stratosphere. The long-range transport of ozone and its precursors outside the modeling domain also contributed to the increase

in MDA8 $O_3$ in China, especially on the Tibetan Plateau (an increase of 1 to 4 ppbv). Our study represents the most comprehensive and up-to-date analysis of the impact of changes in meteorology on ozone across China and highlights the importance of considering meteorological variations when assessing the effectiveness of emission control on changes in the ozone levels in recent years.



## 1 Introduction

Elevated concentrations of ozone ($O_3$) on the earth's surface are harmful to human health and terrestrial vegetation (Lefohn et al., 2018; Lelieveld et al., 2015; Fleming et al., 2018). With rapid urbanization and economic development, ozone pollution has become a major concern in China (Wang et al., 2017; Verstraeten et al., 2015). The magnitude and frequency of high-ozone events are much greater in China than in cities in Japan, South Korea, Europe and the United States (Lu et al., 2018). In 2013, the Chinese government launched the Air Pollution Prevention and Control Action Plan to reduce anthropogenic emissions.

The Chinese Ministry of Ecology and Environment reported that the observed concentrations of primary pollutants had decreased significantly since these strict control measures (http://www.mee.gov.cn). However, the ozone concentrations in major urban areas of China have continued to increase (Lu et al., 2018), and the ozone problem has become a new challenge to air quality management in China. A comprehensive understanding of the causes of the increase in surface ozone levels in China is necessary to develop a comprehensive whole-air improvement strategy.

Ground-level ozone is produced in situ by chemical reactions from ozone precursors, $NO_x$, volatile organic compounds (VOCs), and carbon monoxide (CO) or transported from outside the region and from higher altitudes (Atkinson, 2000; Roelofs and Lelieveld, 1997; Akimoto et al., 2015). Meteorological conditions affect the surface ozone concentrations directly via changes in chemical reaction rates, dilution, wet and dry removal, and transport flux or indirectly via changes in natural emissions (Lu et al., 2019b; Lin et al., 2008). As for the direct effects, an increase in temperature can enhance ozone formation by altering

the chemical reaction rates (Lee et al., 2014; Fu et al., 2015), and an increase in relative humidity can lead to a decrease in ozone concentrations in the lower troposphere (Kalabokas et al., 2015; He et al., 2017). An increase in the planetary boundary layer (PBL) height can decrease ozone levels via dilution of primary pollutants into a larger volume of air (Sanchez-Ccoyllo et al., 2006). Cloud has also been shown to decrease ozone concentrations via aqueous-phase chemistry and photochemistry, which reduces oxidation and enhances cleaning efficiency in the troposphere (Lelieveld and Crutzen, 1990), and precipitation

increases the ozone concentration via the wet removal process (Meleux et al., 2007). Wind fields can significantly affect ozone by transporting ozone and ozone precursors in and out of the region of interest (Lu et al., 2019a; Sanchez-Ccoyllo et al., 2006). As for the indirect effects, an increase in temperature can enhance the biogenic emissions of VOCs and thus affect ozone production (Tarvainen et al., 2005; Guenther et al., 2006; Im et al., 2011).

Several studies have used the statistical analysis or numerical modeling to assess the effects of meteorological variations on

the recent urban ozone trend in China. Using a convergent cross-mapping method to overcome the interactions between various factors, Chen et al. (2019) quantified the influence of individual meteorological factors on the $O_3$ concentration in Beijing from 2006 to 2016. The results indicated that temperature was the critical meteorological driver of the summer ozone concentrations in Beijing. Cheng et al. (2019) applied the Kolmogorov-Zurbenko filtering method to the ozone variations in Beijing from 2006 to 2017, and the results suggested that the relative contribution of meteorological conditions to long-term variation in





ozone was only 2% to 3%, but short-term ozone concentrations were affected significantly by variations in meteorological conditions. Yin et al. (2019) also used the Kolmogorov-Zurbenko approach to analyze the ozone data for Guangzhou from 2014 to 2018 and showed that four meteorological factors, including temperature, relative humidity, etc, accounted for 76% of the variability in the baseline ozone level. Wang et al. (2019b) used a chemistry transport model (Community Multiscale Air Quality modeling system; CMAQ) to investigate the response of summer ozone concentrations to changes in

meteorological conditions from 2013 to 2015, and showed that the maximum daily 8-hour average (MDA8) $O_3$ concentration decreased by 5 to 10 ppb in most cities due to changes in meteorological conditions and biogenic emissions in the latter two years, except for some cities in eastern, south-central, and southwestern China in which the ozone concentration increased by less than 10 ppb. Lu et al. (2019a) used the GEOS-Chem model to explore changes in source attributions contributing to ozone changes over China in 2016 and 2017 and suggested that the increases in ozone in 2017 relative to 2016 were mainly caused

by higher background ozone driven by hotter and drier weather conditions.

Despite these studies, a more comprehensive understanding of the role of the meteorological conditions in the recent ozone changes is still warranted. Previous analyses with a statistical method have been limited to a few cities (i.e., Beijing and Guangzhou). China has a vast territory with a wide range of climates, so the meteorological conditions in various parts of China may have experienced different changes in recent years. Previous chemical transport modeling studies examined either

the meteorological impact for 2 or 3 years (no more), the combined (not individual) effects of meteorological parameters, or the combined (not separate) effect of biogenic emission and meteorology changes.

The objective of our study is to investigate the effects of changes in meteorological conditions and anthropogenic emissions on summer surface ozone increases over China from 2013 to 2017 using an up-to-date regional chemical transport model driven by the interannual meteorological conditions and anthropogenic emissions over the 5 years. This paper (part 1) assesses

the role of meteorological conditions, and a companion paper (part 2) focuses on the role of anthropogenic emissions and implication for multi-pollutant control. Section 2 introduces the observation data, the model used, and experiment settings. In Section 3, we first evaluate the simulated meteorological factors and pollutant concentrations based on the observation data. Subsequently, we separate the changes in the MDA8 $O_3$ concentration due to the variations in meteorological conditions and anthropogenic emissions by conducting numerical sensitivity experiments and explore their contributions to the ozone changes

during the 5 years. Considering the importance of biogenic emissions to ozone production, we estimate the meteorology-driven biogenic emissions over China from 2013 to 2017 and assess their impacts on the variations in ozone. Lastly, the effects of changes in individual meteorological factors are examined, and the role of long-range transport is assessed. Section 4 summarizes the conclusions.


## 2 Methods

### 2.1 Measurement data

We used observation data to evaluate the meteorological parameters and air pollutant concentrations simulated by the Weather Research and Forecasting (WRF)-CMAQ model. The daily meteorological observations were obtained from the National Meteorological Information Center (http://data.cma.cn), including the daily average temperature at a height of 2 m, relative humidity at a height of 2 m, wind speed at a height of 10 m, and surface pressure at ~700 ground weather stations in China. The observed concentrations of air pollutants were obtained from the China National Environmental Monitoring Center (http://106.37.208.233:20035/), including $SO_2$, $NO_2$, CO, MDA8 $O_3$, and $PM_{2.5}$. In 2013, there were 493 environmental monitoring stations in 74 major cities, mostly in urban areas. As a result, only these stations have continuous 5-year observations of pollutants from 2013 to 2017. With the increasing recognition of the air pollution problem in China, more monitoring stations have been built since 2013, and the total number exceeded 1500 in 2017. We applied data quality control to the observed pollutant concentrations to remove unreliable outliers following the approach used in previous studies (Lu et al., 2018; Song et al., 2017). The locations of environmental monitoring stations are presented in Fig. S1.

To evaluate the model performance, we calculated some statistical parameters, including the mean observation, mean simulation, mean bias, mean absolute gross error, root mean square error, index of agreement, and correlation coefficient. The calculation equations of these statistical parameters can be found in Fan et al. (2013).

### 2.2 Model setting

The CMAQ modeling system (Byun and Schere, 2006) was developed by the United States Environmental Protection Agency (US EPA) to approach air quality as a whole by including state-of-the-science capabilities to model multiple air quality issues, including tropospheric ozone, fine particles, toxins, acid deposition, and visibility degradation. This study used the latest CMAQ model (version 5.2.1), an offline chemical transport model without considering the effects of air pollutants on meteorological fields. The meteorological inputs are driven by the WRF model. Table S1 shows the settings of the physical parameterization schemes for the WRF model. The meteorological initial and boundary conditions were provided by NCEP/NCAR FNL reanalysis data with a horizontal resolution of 1°. Fig. S1 shows the modeling domains for the WRF and CMAQ model with a horizontal resolution of 36 km. The model has 23 vertical layers and reaches 50 hPa. The CMAQ modeling domain covers all the land area of China and the surrounding regions, which is a few grids smaller than the WRF modeling domain to reduce the effect of the meteorological boundary from the WRF model. The boundary conditions of chemical species for CMAQ were derived from the modeling results of the global chemistry transport model, Model for Ozone and Related Chemical Tracers, version 4 (MOZART-4) (http://www.acom.ucar.edu/wrf-chem/mozart.shtml) (Emmons et al., 2010). We used SAPRC07TIC (Carter, 2010; Hutzell et al., 2012; Xie et al., 2013; Lin et al., 2013) as the gas-phase chemical mechanism and AERO6i (Murphy et al., 2017; Pye et al., 2017) as the aerosol mechanism in the CMAQ model.



Aerosol also has a significant effect on ozone via changes in the photolysis rate and heterogeneous reactions (Lou et al., 2014; Liao and Seinfeld, 2005). The original CMAQ model includes only the heterogeneous reactions of $NO_2$, $NO_3$, and $N_2O_5$ on aerosol surfaces. To better simulate the effects of aerosol on ozone via heterogeneous reactions, we updated the heterogeneous reaction rates of $NO_2$ and $NO_3$ on the aerosol surface and incorporated more heterogeneous reactions into the CMAQ model, including the uptake of $HO_2$, $O_3$, OH, and $H_2O_2$. The detailed heterogeneous reactions in the updated CMAQ model are listed

in Table S2. We select the "best guess" uptake coefficients of these gases, which have been widely used in previous chemical transport model studies (Jacob, 2000; Zhu et al., 2010; Zhang and Carmichael, 1999; Fu et al., 2019; Liao et al., 2004). These improvements help the CMAQ model better simulate ozone and other pollutants, and their influence and that of aerosol on the ozone concentration via various heterogeneous reactions are evaluated in the companion paper (part 2).

### 2.3 Emission

For anthropogenic emissions, we used the multi-resolution emission inventory for China (MEIC) for 2013 to 2017 (http://www.meicmodel.org/), which was developed by Tsinghua University and has been evaluated by satellite data and ground observation (Zheng et al., 2018). International shipping emissions in 2010 were obtained from the Hemispheric Transport Atmospheric Pollution emissions version 2.0 dataset (Janssens-Maenhout et al., 2015). Biogenic emissions from 2013 to 2017 were calculated from the Model of Emissions of Gas and Aerosols from Nature (MEGAN) (Guenther et al., 2006)

driven by the interannual summer meteorological inputs from the WRF model.

### 2.4 Experiment setting

The model simulations were conducted for the summers (June, July, and August) from 2013 to 2017, driven by interannual meteorology and anthropogenic emissions, namely, the base simulations. The shipping emissions remained unchanged in the 5-year simulation due to a lack of data for recent years. To investigate the causes of the increasing surface ozone levels in

China, we established four sets of modeling experiments based on the simulation of 2013. The first was designed to evaluate the effects of changes in meteorological conditions and anthropogenic emissions (Table S3). We derived the effects of meteorological variation by comparing the simulated ozone concentrations in different years but with the same anthropogenic emissions and chemical boundary conditions as those from 2013. The effects of changes in anthropogenic emissions were derived by comparing the simulated ozone concentrations in 2013 but with anthropogenic emissions from different years. The

second set was designed to evaluate the effects of variations in biogenic emissions driven by meteorological conditions (Table S4), which were derived by comparing the simulated ozone concentrations in 2013 but with biogenic emissions from different years. The third set was designed to evaluate the contributions of the individual meteorological parameters to the ozone change from 2013 to 2017 (Table S5), including temperature, specific humidity, wind field, PBL height, clouds, and precipitation. The fourth set was designed to evaluate the contribution of long-range transport from outside the modeling domain (Table S6) by



comparing the simulated ozone concentrations in 2013 with those with chemical boundary conditions from MOZART from

different years.

### 3 Results

### 3.1 Model evaluation

Table 1 shows the evaluation results for temperature, relative humidity, wind speed, and surface pressure. The results for all

weather stations in China were averaged. The simulated temperature at a height of 2 m was slightly underestimated with biases

of less than 0.6 °C in 5 years. The high correlation coefficients (over 0.82) indicate that the WRF model can capture variations

in temperature. Like the temperature, the simulated relative humidity was also slightly under-predicted and had a high

correlation coefficient with the observation. The simulated wind speed at a height of 10 m was slightly overestimated by about

0.5 m s$^{-1}$ due to the underestimation of the effects of urban topography in the WRF model and was often found in other WRF

modeling studies (Fan et al., 2015; Hu et al., 2016). The WRF model faithfully reproduces surface pressure for 5 years with

low biases and high correlation coefficients. The WRF model could also capture the temporal variations in meteorological

parameters. For example, the simulated temperature at a height of 2 m decreased from 2013 to 2015 and then increased from

2015 to 2017, which is consistent with the observations. The good performance of the WRF model gives us the confidence to

use the simulations to study the effects of variations in meteorological conditions on the ozone level.

Table 2 presents the evaluation results for air pollutant concentrations in China. Generally, the CMAQ model has excellent

performance on the simulated pollutant concentrations with low biases, high index of agreement, and high correlation

coefficients. The simulated $NO_2$ concentration was slightly underestimated for these 5 years in general, which can be explained

in part by the fact that the $NO_2$ concentrations in the national network were measured using the catalytic conversion method,

which overestimates $NO_2$, especially during periods with active photochemistry and at locations away from primary emission

sources (Xu et al., 2013; Zhang et al., 2017; Fu et al., 2019). The simulated CO concentration is underestimated significantly

by the CMAQ model, which might be due to the missing sources of CO such as biomass burning. The CMAQ model predicts

a slightly higher MDA8 $O_3$ concentration, which could be explained by the artificial mixing of ozone precursors in modeling

grids leading to higher ozone production efficiency and positive ozone biases, especially for models with coarser resolutions

(Young et al., 2018; Chen et al., 2018; Yu et al., 2016). However, the overall CMAQ model performance is acceptable and can

support further investigation of the drivers of increasing ozone levels in China.

### 3.2 Rate of change in ozone due to meteorology and anthropogenic emission

Fig. 1 shows the spatial distribution of the summer surface MDA8 $O_3$ level over China in summer from 2013 to 2017. The

CMAQ model can faithfully capture the spatiotemporal variations in the observed MDA8 $O_3$ level. Both the simulations and

the observations exhibit elevated concentration in midlatitude areas, including the North China Plain (NCP), Yangtze River



Delta (YRD), Sichuan Basin (SCB), and large areas in central and western China. The concentrations in southern China are

lower than those in northern China, but they are relatively high in the Pearl River Delta (PRD) region.

We applied the linear regression method to obtain rates of change in the interannual MDA8 $O_3$ concentration for simulation

and observation, which are shown in Fig. 2. In general, the observed MDA8 $O_3$ concentrations present an increasing trend in

most monitoring stations (most located in urban areas) of China from 2013 to 2017. With the aid of the model simulations, the

characteristics of the changes in MDA8 $O_3$ levels were revealed for all areas, including those with no monitoring stations. Both

observations and model simulations show that NCP, YRD, SCB, northeastern China, and some areas in western China

experienced increasing levels of ozone pollution. Interestingly, the model results revealed that MDA8 $O_3$ levels were

decreasing in large parts of rural areas that could not be covered by the current monitoring stations, such as northwestern China

and southern China.

We separated the change rates of simulated MDA8 $O_3$ into that due to variations in meteorological conditions and changes in

anthropogenic emissions (also see Fig. 2). Here, the impact of biogenic emission variation is included in the effects of

meteorological variation because it is affected by meteorology. The result shows that the change rates of ozone over China

were more affected by meteorological changes than by emission changes in terms of spatial distribution. The regions with an

increasing or decreasing trend of ozone were generally consistent with the contributions from variations in meteorology except

for some regions whose ozone trends were dominated by anthropogenic emission changes. The changes in anthropogenic

emission have resulted in ozone increases in NCP, YRD, PRD, SCB, and other scattered megacities but decreases in rural

regions. This discrepancy can be explained by the different ozone formation regimes in urban (VOCs-limited) and rural ($NO_x$-

limited) areas (Li et al., 2019a; Wang et al., 2019a). A recent study reported the observations of surface ozone during 1994-

2018 at a coastal site in southern China and revealed no significant changes in the ozone levels in the outflow of air masses

from China mainland during the recent years (Wang et al., 2019c). These results suggest that nationwide $NO_x$ emission

reductions may have decreased ozone production over large regions despite causing ozone increase in urban areas. The impact

of anthropogenic emission changes on ozone levels in recent years remains a challenging and momentous topic and will be

assessed in the companion paper (part 2). This paper focuses on the effects of meteorological conditions.

**3.3 Impact of meteorological conditions and anthropogenic emissions relative to 2013**

We next quantified the impact of meteorological conditions and anthropogenic emissions to ozone changes from 2013 to 2017

relative to 2013 (Fig. 3). The changes in MDA8 $O_3$ from the base simulation varied spatially and yearly, mainly as a result of

meteorological conditions. The variation in the MDA8 $O_3$ concentration due to meteorological changes ranged from -12.7 to

15.3 ppbv over China from 2014 to 2017 relative to 2013. The emission-induced MDA8 $O_3$ changes in each year exhibited

similar spatial patterns, which were consistent with those of the change rates due to emission changes (Fig. 2d). The impact of

emission changes on the MDA8 $O_3$ concentrations became increasingly significant as anthropogenic emissions were further



reduced. Our results differ from those by Wang et al. (2019b) which suggested a less important role of meteorology in the variation of ozone from 2013 to 2015. The discrepancy could be due to the difference in the chemical mechanisms and method used for quantifying the effects of emission changes. They calculated the changes in ozone levels due to emission variations by subtracting simulated changes due to meteorological conditions variations from total observed changes, and we calculated

by comparing the simulated difference between the simulations in 2013 driven by anthropogenic emissions from different years.

We further found that in some specific regions and years, the changes in MDA8 $O_3$ concentrations due to meteorological variation could be comparable with or greater than those due to emission changes, which highlights the significant role of meteorological conditions in ozone variations. As a result, we selected four megacities in different regions of China to further

examine the impact of changes in meteorological conditions and anthropogenic emissions on ozone levels (Fig. 4)—Beijing, Shanghai, Guangzhou, and Chengdu—which are principal cities in the Beijing-Tianjin-Hebei (BTH) region in the north, the YRD in the east, the PRD in the south, and the SCB in the southwestern part of China, respectively (see Fig. S1 for their locations). The numbers of monitoring sites used to obtain the average values for these four cities were 12, 9, 11, and 8, respectively. Most of these sites are situated in the city center, and thus the average results represent the conditions in urban

areas. As shown in Fig. 4, the model can generally capture variations in MDA8 $O_3$ in these cities (except for the changes of MDA8 $O_3$ in Beijing in 2014 and 2015, which were underestimated, likely due to underestimation of anthropogenic emissions in Beijing and its surrounding regions). The contributions of anthropogenic emissions to MDA8 $O_3$ exhibited an almost linear increasing trend in the four cities from 2013 to 2017, whereas the contribution of meteorology could be positive or negative, depending on the region and year.

In Beijing (the BTH region), the variations in meteorological conditions had little effect on the MDA8 $O_3$ changes from 2014 to 2017 relative to 2013. The increase in ozone was driven primarily by the changes in anthropogenic emissions. This characteristic can also be identified in the larger BTH region, as shown in Fig. 3. In Shanghai (the YRD region), the effects of meteorology were comparable with those of anthropogenic emissions in terms of the absolute values of the contribution to MDA8 $O_3$ changes. From 2014 to 2016, the meteorology was unfavorable for ozone formation, which masks the ozone increase

due to emission changes. However, meteorological conditions became a positive driver in 2017, leading to a drastic increase in the total MDA8 $O_3$ concentration (over 10 ppb). In Guangzhou (the PRD region), the role of meteorological conditions was opposite that in Shanghai. The weather changes were conducive to ozone formation from 2014 to 2016 compared with 2013, contributing to a large increase in MDA8 $O_3$ by over 10 ppbv; in 2017, however, the impact of changes in meteorological conditions on ozone levels decreased substantially, leading to a moderate increase in MDA8 $O_3$ in that year compared with

2013. In Chengdu (the SCB region), the impact of meteorological conditions on ozone variation was limited in these years, and the ozone concentration was mainly affected by emission changes, similar to the situation in Beijing. Our result is similar to those by Wang et al. (2019b) for Shanghai and Guangzhou from 2013 to 2015, both indicating meteorological variations


unfavorable for ozone formation in Shanghai and favorable in Guangzhou. On the other hand, our simulations differ from theirs that showed a considerable negative contribution of the meteorology variations to ozone levels in Beijing and Chengdu.

Our study and that of Chen et al. (2019) suggest a weak role of meteorology variation in the summer ozone trend in Beijing. In addition to these four regions, we found a significant impact of meteorology on the ozone change in western China, especially the Tibetan Plateau (Fig. 3). The meteorological variations contributed to considerable increases in MDA8 $O_3$ in 2014-2017 in this region relative to 2013.

**3.4 Impact of meteorology-driven biogenic emissions relative to 2013**

Temperature is an important meteorological driver of biogenic emissions. Fig. 5 displays the temperature changes in China from 2014 to 2017 compared with 2013. The changes in the spatial distribution were similar in these four years. A decrease in temperature was found in midlatitude areas (25°N to 40°N), and an increase was found in southern (south of 25°N) and northern China (north of 40°N). As shown in Fig. S3, in midlatitude areas such as the BTH, YRD, and SCB, the temperature decreased from 2013 to 2015 and then increased from 2015 to 2017. In contrast, in southern China, such as the PRD region, the

temperature increased during 2013-2014 and then slightly decreased during 2014-2017. The variation in the observed temperature is well captured by the WRF model, which enables the MEGAN model to calculate the variation of biogenic emissions driven by the realistic temperature. We present the results of biogenic isoprene emissions because isoprene is generally the most abundant biogenic VOC and has the highest ozone formation potential (Zheng et al., 2009). Large isoprene emissions were found in the southern parts of China and northeast China, which have high vegetation covers in summer (Fig.

5f). The spatial and interannual variations of biogenic isoprene emissions followed the changes in temperatures in China, leading to similar changes in MDA8 $O_3$ concentrations.

In midlatitude areas of China, the variations in biogenic emissions induced a decrease in the MDA8 $O_3$ concentration after 2013. The most significant decrease in the MDA8 $O_3$ concentration was found in the YRD and SCB regions, where there were high biogenic emissions and a drastic temperature decrease. In 2014 and 2015, the MDA8 $O_3$ concentration decreased by ~5

ppbv in these two regions compared with 2013. The changes in ozone concentration were less affected by biogenic emissions due to the lower biogenic emissions and smaller variation of temperature in the BTH region. In southern and northern China, the increase in temperature and then biogenic emissions since 2013 led to an enhancement of the MDA8 $O_3$ concentration by up to 1 to 2 ppbv (Fig. 5). In Guangzhou, for example, affected by temperature-dependent biogenic emissions, the MDA8 $O_3$ concentration increased by 0.8 ppbv from 2013 to 2014 and then decreased slightly from 2014 to 2017 (Fig. S4).

The changes in MDA8 $O_3$ concentrations due to changes in biogenic emissions in Shanghai and Guangzhou (Fig. S4) generally matched the total changes in ozone levels due to variations in meteorological conditions and provided a considerable contribution to them (Fig. 4). The variations in biogenic emissions were mostly affected by temperature. In section 3.5, we also found that the changes in ozone levels caused by temperature variations via altering chemical reaction rates had an even





more significant impact than via changing biogenic emissions in 2017 relative to 2013. As a result, the temperature can play

an important role in the variations in ozone levels in recent years. Previous studies also demonstrated the significant role of

temperature in the ozone trend in China and other regions (Hsu, 2007; Jing et al., 2014; Lee et al., 2014). However, the role of

meteorology is complex and the changes in other meteorological factors can counteract this effect. In Beijing and Chengdu,

for example, the changes in ozone levels due to variations in meteorology were insignificant and could not reflect those caused

by variations in temperature-dependent biogenic emissions (Fig. 4).

**3.5 Impact of individual meteorological parameters in 2017 relative to 2013**

Fig. 6 shows the individual effects of changing temperature, humidity, wind field, PBL height, clouds, and precipitation

between 2017 and 2013 on the ozone level. Of all the meteorological parameters, the change of wind fields had the most

significant impact on the MDA8 $O_3$ concentration. It led to an increase in MDA8 $O_3$ in nearly all of China, with a maximum

of 9.1 ppbv (Fig. 6i). In western and central China, notable increases in the MDA8 $O_3$ concentrations due to the change in

wind fields were identified, which contributed significantly to the meteorology-induced increasing ozone (Fig. 3h). This

concentration increase due to changes in the wind fields cannot be caused by horizontal transport because we could not find

regions with a significant decrease in ozone over China. We also could not find a significant relationship between the changes

in wind speed and ozone concentrations. Thus, the increase in vertical transport to the surface is believed to be the leading

cause. An examination of the potential velocity in the upper troposphere shows that the tropopause height decreased in central

China from 2013 to 2017 (Fig. 6h), and more ozone was transported from the stratosphere to the troposphere via stratosphere-

troposphere exchange. Our model simulations show the increase in MDA8 $O_3$ by 3 to 9 ppbv between 2013 and 2017 due to

wind field changes in western China, with the largest increase over the Tibetan Plateau. The simulated potential velocities from

2014 to 2017 have increased since 2013 and exhibited a similar pattern of change (Fig. S5). These changes were also revealed

by the potential velocity reanalysis data from the European Centre for Medium-Range Weather Forecasts (ECMWF,

https://apps.ecmwf.int/datasets/). It indicates that the increased transport of ozone from the lower stratosphere contributed to

the increasing surface ozone in China in recent years.

In addition to wind fields, other meteorological parameters contribute to the ozone change. Because a high temperature

facilitates the formation of ozone via the increase in chemical reaction rates, the changes in ozone due to temperature were

consistent with the changes in temperature in terms of spatial distribution (Fig. 6b and c). The MDA8 $O_3$ concentration

decreased in central China and increased in other parts of China. This change in the spatial distribution was similar to that due

to the changes in biogenic emissions because they were both affected by temperature. However, comparing Fig. 5o to Fig. 6c,

we found that the impact of temperature via the change in the chemical reaction rate was more significant than that via the

change in biogenic emissions from 2013 to 2017.

The specific humidity decreased in central China and northeast China but increased in other parts of China from 2013 to 2017



(Fig. 6e). A decrease in the specific humidity in central China led to an increase in the MDA8 $O_3$ concentration in localized

areas, and an increase in other parts of China resulted in a decrease in the MDA8 $O_3$ concentration in a large area (Fig. 6f). A

negative correlation between ozone concentration and humidity in various regions over China was also reported in many

previous studies (Ma et al., 2019; Li et al., 2019b).

From 2013 to 2017, the PBL height increased in most parts of China, including NCP, northeast China, and northwest China

(Fig. 6k). Our modeling results show that the increase in the PBL height enhanced the MDA8 $O_3$ concentration in most parts

of China (Fig. 6l). A positive correlation between the PBL height and the ozone level in China is also revealed in the statistical

results of He et al. (2017). Possible reasons for the ozone increase with the increase in the PBL height include low primary

pollutant concentrations with the development of PBL (He et al., 2017) and downward transport of ozone from the upper

troposphere where ozone is higher than the near-surface (Sun et al., 2009).

The cloud fraction increased in southwestern China and the Tibetan Plateau but slightly decreased in other parts of China from

2013 to 2017 (Fig. 6n). Because clouds can decrease ozone concentrations via aqueous-phase chemistry and photochemistry

to reduce oxidation and enhance cleaning efficiency in the troposphere (Lelieveld and Crutzen, 1990), the MDA8 $O_3$

concentration in most parts of China increased except for southwestern China and the Tibetan Plateau (Fig. 6o).

The change in precipitation was similar to that of the cloud fraction in terms of spatial distribution (Fig. 6q) but made an

opposite contribution to ozone concentrations compared with clouds (Fig. 6r). A positive correlation (p=0.95) between ozone

and precipitation was also reported by the statistical results of Li et al. (2019b). Although precipitation can remove ozone from

the atmosphere (Meleux et al., 2007), an increase in precipitation may decrease aerosol concentrations that could increase

ozone levels by altering photolysis rates and heterogeneous reactions.

The meteorological parameters that dominate ozone changes can differ in the four megacities (Fig. S6). The     decrease    in

cloud cover was the important meteorological cause that increased the MDA8 $O_3$ concentration in Beijing from 2013 to 2017,

the wind field change was the dominant factor that increased and decreased the MDA8 $O_3$ concentration in Shanghai and

Guangzhou, respectively, and the decrease in temperature contributed primarily to the decline in the MDA8 $O_3$ concentration

in Chengdu. However, the effect of the dominant meteorological factor on variations in the ozone level could be counteracted

by the influence of other meteorological factors. For example, in Shanghai, the significant positive effect of changes in wind

fields on ozone formation was offset by the negative effects of changes in temperature and precipitation, leading to the smaller

increase in the ozone level due to the overall meteorological changes in 2017 compared with 2013 (Fig. 6b). The synergistic

or counteracting effects from individual meteorological factors can give rise to the complex impact of the overall meteorology

on ozone variations.

**3.6 Impact of long-range transport relative to 2013**

The chemical boundary conditions for the CMAQ model were derived from the results of the MOZART global model, which





can represent the long-range transport of ozone and its precursors into China. We changed the chemical boundary conditions in 2013 to different years to investigate the role of long-range transport in ozone variations in China, and the results are shown in Fig. 7. Changes in long-range transport after 2013 increased the MDA8 $O_3$ concentration over China except for some areas in northwestern China. Compared with a small increase in MDA8 $O_3$ (<1ppbv) in eastern China, the Tibetan Plateau

encountered a significant increase in the MDA8 $O_3$ concentration by about 1 to 4 ppbv due to changes in long-range transport after 2013. A previous MOZART study by Li et al. (2014) found that the transport from the emissions of all Eurasian regions except China contributed 10 to 15 ppbv to the surface $O_3$ concentration over western China. Peroxyacetyl nitrate can be transported a long distance in the cold free troposphere and then be thermally decomposed to release $NO_x$, leading to the enhancement of ozone formation with high efficiency in remote regions (Fischer et al., 2014). An analysis of 10-day back

trajectories at Mount Waliguan (a remote mountain site in western China) also showed that the air mass group from central Asia contributed to the high $O_3$ levels observed at the site during summer via long-range transport in the free troposphere (Wang et al., 2006). Our study indicates a considerable increase (1 to 4 ppbv) in this long-range transport contribution since 2013.

## 4 Conclusions

This study explored the impact of changes in meteorological conditions and anthropogenic emissions on the recent ozone variations across China. The changes in anthropogenic emissions since 2013 increased the MDA8 $O_3$ concentrations in urban areas but decreased the ozone levels in rural areas. The meteorological impact on the ozone trend varied by region and by year and could be comparable with or even larger than the impact of changes in anthropogenic emissions. The individual effects of changes in temperature, specific humidity, wind field, planetary boundary layer height, clouds, and precipitation from 2013 to

2017 on the ozone levels were examined in this study. The results show that the changes in the wind fields made a significant contribution to the increase in surface ozone levels over China by transporting ozone from the upper troposphere and lower stratosphere. The main findings for various regions of China are summarized as follows.

1) In Beijing (the Beijing-Tianjin-Hebei region), the contribution of meteorological changes to the variations in summer ozone was small in 2014-2017 relative to 2013, and the changes in anthropogenic emissions dominated the increase in

360        ozone. Decreasing cloud cover was the dominant meteorological factor that contributed to the increased concentrations of MDA8 $O_3$.

2) In Shanghai (the Yangtze River Delta region), meteorological variation decreased ozone formation from 2014 to 2016, which masked a large increase in ozone due to changes in emissions. The meteorological conditions in 2017 became a positive driver for an increase in ozone, leading to a drastic increase in the total MDA8 $O_3$ concentration (over 10 ppb).

365        Changes in biogenic emissions had a significant impact on the ozone level in this region. The temperature decreased after



2013 resulted in a considerable decline in the concentrations of MDA8 $O_3$, especially in 2014 and 2015. The wind field change was the dominant factor that increased the MDA8 $O_3$ concentration in 2017 relative to 2013.

3) In Chengdu (the Sichuan Basin region), the impact of meteorological conditions on changes in ozone was limited from 2013 to 2017, and the ozone concentration was mainly affected by emissions, as in Beijing. However, the biogenic emissions induced by meteorological conditions were important and led to a moderate decrease in the ozone concentration. The drop in temperature contributed to the decrease in the MDA8 $O_3$ concentration in 2017 relative to 2013.

4) In Guangzhou (the Pearl River Delta region), the meteorological conditions were more conducive to ozone formation from 2014 to 2016 than in 2013, which led to a significant increase (>10 ppbv) in MDA8 $O_3$. In 2017, the impact of changes in meteorological conditions on ozone levels decreased substantially, and the increase in MDA8 $O_3$ was small and only due to the changes in emissions. The biogenic emissions driven by meteorological conditions were also important to ozone formation and increased the MDA8 $O_3$ concentration after 2013. The wind field change was the dominant meteorological factor that decreased the MDA8 $O_3$ concentration in 2017 compared with 2013.

5) In western China, the increase in ozone concentrations was mainly caused by meteorological conditions. The increase in the MDA8 $O_3$ concentration from 2013 to 2017 was primarily ascribed to enhanced downward transport from the upper troposphere and lower stratosphere. The long-range transport of ozone and its precursors in the upper atmosphere outside the modeling domain also contributed to the increase in MDA8 $O_3$ in most parts of China, especially on the Tibetan Plateau.

In conclusion, our study highlights the complex but varying effects of meteorological conditions on surface ozone levels across the regions of China and for different years. It is therefore necessary to take into consideration of the variations in meteorological conditions when assessing the effectiveness of emission control policies on changes in the levels of ozone (and other air pollutants) in different cities and/or regions of China. Developing future air pollution mitigation policies should also account for the counteracting effect of meteorological variations.

**Author contribution**

T.W. initiated the research, Y.M.L. and T.W. designed the paper framework. Y.M.L. ran the model, processed the data, and made the plots. Y.M.L. and T.W. wrote the paper.

**Competing interests**

The authors declare that they have no conflict of interest.

**Code/Data availability**

The code or data used in this study are available upon request from Yiming Liu (yming.liu@polyu.edu.hk) and Tao Wang



(cetwang@polyu.edu.hk).

**Acknowledgements**

This work was supported by the Hong Kong Research Grants Council (T24-504/17-N) and the National Natural Science Foundation of China (91844301). We would like to thank Prof. Qiang Zhang from Tsinghua University for providing the emission inventory, Prof. Qi Fan from Sun Yat-sen University for help accessing to the meteorological data, Dr. Xiao Lu from Harvard University for sharing consolidated air pollutant observation data, Dr. Xiao Fu from the Hong Kong Polytechnic

University for sharing the model codes of HONO sources.

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





**Table 1: Evaluation results for the meteorological factors (T2 is temperature at a height of 2 m; RH2 is relative humidity at a height of 2 m; WS10 is wind speed at a height of 10 m; PRS is surface pressure; Num is number of sites with available observation for statistics; OBS is mean observation; SIM is mean simulation; MB is mean bias; MAGE is mean absolute gross error; RMSE is root mean square error; IOA is index of agreement; r is correlation coefficient; OBS, SIM, MB, MAGE, and RMSE have the same units as given in the first column, while IOA and r have no unit).**

| Species | Year | Num | OBS | SIM | MB | MAGE | RMSE | IOA | r |
|---|---|---|---|---|---|---|---|---|---|
| T2 | 2013 | 692 | 23.9 | 23.3 | -0.6 | 2.1 | 2.3 | 0.99 | 0.82 |
| (°C) | 2014 | 690 | 23.1 | 22.8 | -0.3 | 1.9 | 2.2 | 0.99 | 0.82 |
| | 2015 | 708 | 23.0 | 22.5 | -0.5 | 1.9 | 2.2 | 0.99 | 0.83 |
| | 2016 | 694 | 23.8 | 23.2 | -0.6 | 2.0 | 2.3 | 0.99 | 0.82 |
| | 2017 | 694 | 23.7 | 23.2 | -0.5 | 2.0 | 2.2 | 0.99 | 0.86 |
| RH2 | 2013 | 692 | 70.5 | 67.6 | -2.9 | 10.1 | 11.9 | 0.99 | 0.72 |
| (%) | 2014 | 690 | 72.2 | 67.2 | -5.0 | 10.1 | 11.9 | 0.99 | 0.68 |
| | 2015 | 708 | 71.2 | 67.6 | -3.6 | 9.2 | 10.9 | 0.99 | 0.72 |
| | 2016 | 694 | 72.0 | 68.4 | -3.6 | 9.2 | 10.9 | 0.99 | 0.71 |
| | 2017 | 694 | 71.7 | 67.7 | -4.1 | 9.4 | 11.1 | 0.99 | 0.73 |
| WS10 | 2013 | 692 | 2.1 | 2.8 | 0.7 | 1.0 | 1.2 | 0.93 | 0.53 |
| (m/s) | 2014 | 690 | 1.9 | 2.5 | 0.5 | 0.9 | 1.0 | 0.94 | 0.47 |
| | 2015 | 708 | 2.1 | 2.6 | 0.6 | 0.9 | 1.1 | 0.94 | 0.54 |
| | 2016 | 694 | 2.1 | 2.6 | 0.5 | 0.9 | 1.1 | 0.94 | 0.50 |
| | 2017 | 694 | 2.1 | 2.6 | 0.5 | 0.9 | 1.1 | 0.94 | 0.49 |
| PRS | 2013 | 692 | 922.4 | 906.2 | -16.2 | 21.0 | 21.0 | 0.99 | 0.98 |
| (hPa) | 2014 | 690 | 924.0 | 907.8 | -16.3 | 21.1 | 21.1 | 0.99 | 0.98 |
| | 2015 | 708 | 924.0 | 907.9 | -16.1 | 21.1 | 21.1 | 0.99 | 0.97 |
| | 2016 | 694 | 923.4 | 907.5 | -15.9 | 20.7 | 20.7 | 0.99 | 0.98 |
| | 2017 | 694 | 923.4 | 907.8 | -15.6 | 20.5 | 20.6 | 0.99 | 0.98 |





**Table 2: Evaluation results for the air pollutant concentrations in China (Num is number of sites with available observation for statistics; OBS is mean observation; SIM is mean simulation; MB is mean bias; MAGE is mean absolute gross error; RMSE is root mean square error; IOA is index of agreement; r is correlation coefficient; OBS, SIM, MB, MAGE, and RMSE have the same units as given in the first column, while IOA and r have no unit).**

| Species | Year | Num | OBS | SIM | MB | MAGE | RMSE | IOA | r |
|---|---|---|---|---|---|---|---|---|---|
| $SO_2$ | 2013 | 408 | 7.1 | 12.0 | 4.9 | 7.6 | 9.1 | 0.79 | 0.28 |
| (ppbv) | 2014 | 867 | 6.4 | 9.0 | 2.6 | 5.9 | 7.0 | 0.80 | 0.26 |
| | 2015 | 1410 | 5.0 | 5.2 | 0.2 | 4.0 | 4.8 | 0.77 | 0.23 |
| | 2016 | 1422 | 4.4 | 4.1 | -0.3 | 3.4 | 4.0 | 0.77 | 0.24 |
| | 2017 | 1474 | 3.8 | 3.2 | -0.6 | 2.7 | 3.1 | 0.77 | 0.22 |
| $NO_2$ | 2013 | 430 | 15.1 | 16.6 | 1.4 | 7.0 | 8.3 | 0.91 | 0.41 |
| (ppbv) | 2014 | 843 | 13.9 | 13.8 | -0.1 | 6.6 | 7.7 | 0.89 | 0.37 |
| | 2015 | 1411 | 11.3 | 9.9 | -1.4 | 5.7 | 6.7 | 0.84 | 0.34 |
| | 2016 | 1420 | 10.9 | 9.5 | -1.4 | 5.5 | 6.4 | 0.85 | 0.35 |
| | 2017 | 1480 | 11.3 | 9.5 | -1.8 | 5.9 | 6.8 | 0.83 | 0.32 |
| CO | 2013 | 436 | 0.71 | 0.34 | -0.37 | 0.39 | 0.45 | 0.81 | 0.33 |
| (ppmv) | 2014 | 872 | 0.75 | 0.32 | -0.42 | 0.44 | 0.49 | 0.79 | 0.34 |
| | 2015 | 1400 | 0.65 | 0.28 | -0.38 | 0.39 | 0.44 | 0.78 | 0.32 |
| | 2016 | 1419 | 0.65 | 0.26 | -0.39 | 0.40 | 0.44 | 0.78 | 0.33 |
| | 2017 | 1473 | 0.62 | 0.25 | -0.37 | 0.38 | 0.41 | 0.78 | 0.30 |
| MDA8 $O_3$ | 2013 | 371 | 50.9 | 57.7 | 6.8 | 17.8 | 21.3 | 0.95 | 0.55 |
| (ppbv) | 2014 | 836 | 52.5 | 59.2 | 6.7 | 17.7 | 21.1 | 0.95 | 0.54 |
| | 2015 | 1361 | 50.4 | 56.4 | 5.9 | 15.3 | 18.3 | 0.96 | 0.55 |
| | 2016 | 1373 | 52.3 | 57.6 | 5.3 | 13.4 | 16.3 | 0.97 | 0.61 |
| | 2017 | 1440 | 56.3 | 58.3 | 1.9 | 13.1 | 16.1 | 0.98 | 0.63 |
| $PM_{2.5}$ | 2013 | 437 | 44.4 | 42.8 | -1.7 | 19.4 | 26.0 | 0.91 | 0.58 |
| ($\mu g/m^3$) | 2014 | 869 | 43.8 | 43.6 | -0.2 | 19.1 | 24.5 | 0.92 | 0.57 |
| | 2015 | 1401 | 35.3 | 31.6 | -3.7 | 16.4 | 20.6 | 0.89 | 0.54 |
| | 2016 | 1411 | 29.7 | 27.0 | -2.7 | 13.5 | 17.0 | 0.90 | 0.54 |
| | 2017 | 1462 | 27.8 | 24.5 | -3.3 | 12.6 | 15.8 | 0.89 | 0.52 |






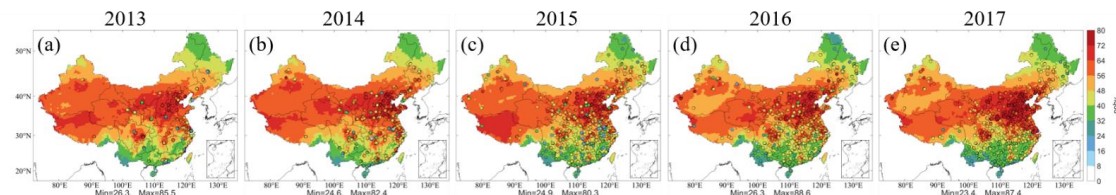

**Figure 1: Spatial distribution of the simulated surface maximum daily 8-hour average (MDA8) O$_3$ concentration across China in summer (June-August) of 2013 (a), 2014 (b), 2015 (c), 2016 (d), and 2017 (e). Circles with color are the available observed values at environmental monitoring stations in each year.**

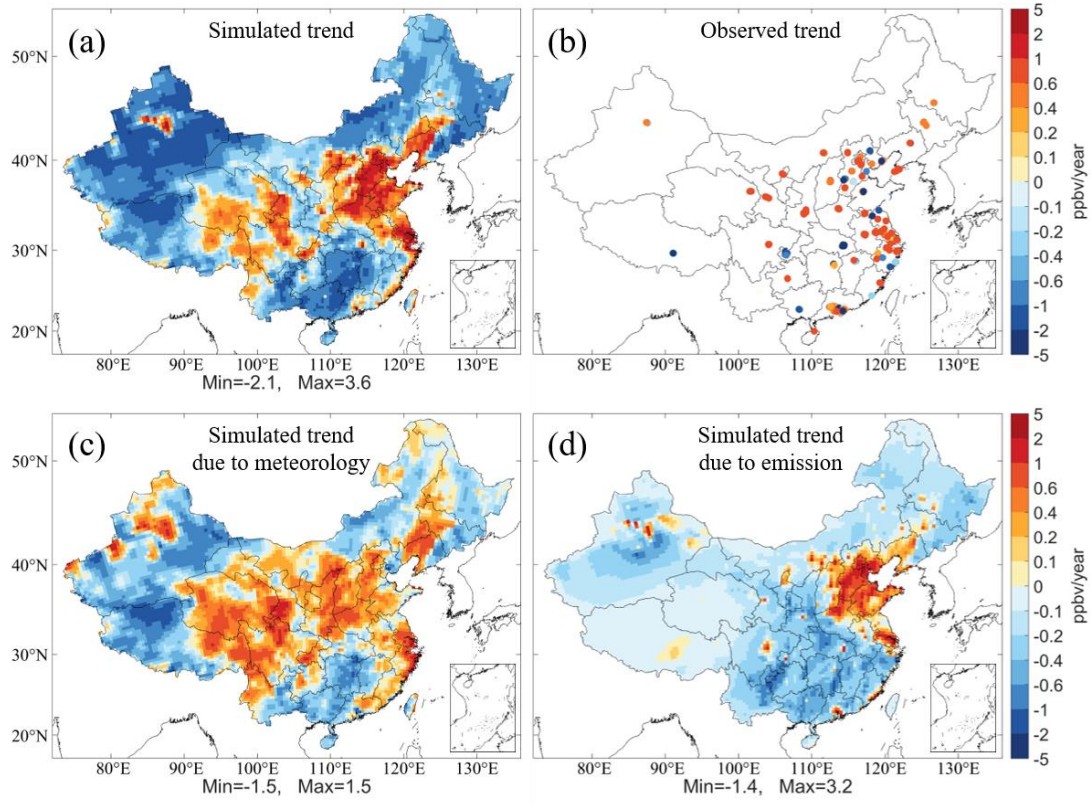


**Figure 2: Rates of changes in the simulated (a) and observed (b) surface MDA8 O$_3$ concentrations over China in summer from 2013 to 2017. As for the observation, only environmental monitoring sites (493) with data available in all 5 years are presented. (c) and (d) present the rates of changes in the simulated MDA8 O$_3$ concentrations due to variations in meteorological conditions and anthropogenic emissions over China in summer from 2013 to 2017 (see methods). The corresponding p values of regression are**

**presented in Fig. S2.**


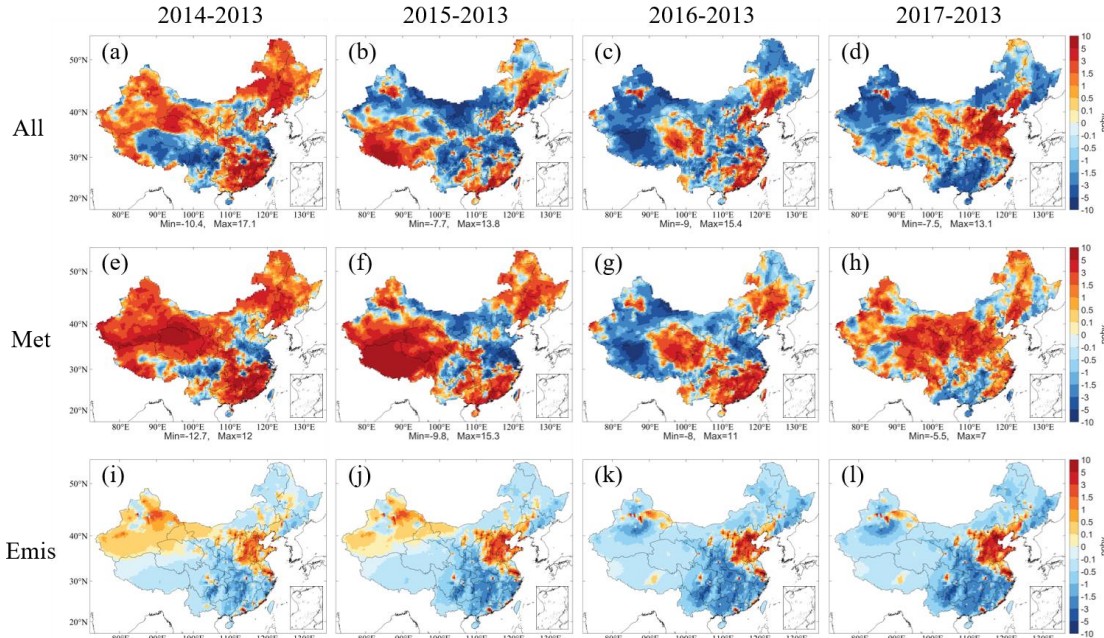

**Figure 3: Changes in the simulated summer surface MDA8 O₃ concentrations from the base simulation (All, the top row), and those due to variations in meteorological conditions (Met, the central row) and anthropogenic emissions (Emis, the bottom row) in 2014, 2015, 2016 and 2017 relative to 2013.**



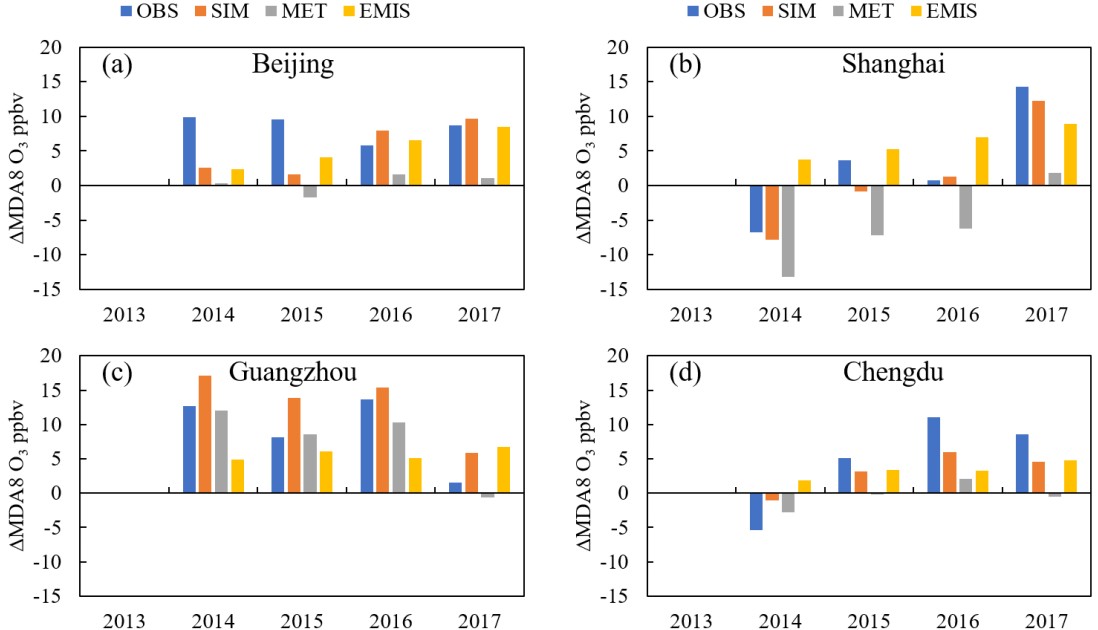

**Figure 4: Interannual changes in the simulated (SIM) and observed (OBS) summer surface MDA8 O₃ concentrations and those due to variations in meteorological conditions (MET) and anthropogenic emissions (EMIS) in (a) Beijing, (b) Shanghai, (c) Guangzhou, and (d) Chengdu in 2013-2017 relative to 2013.**

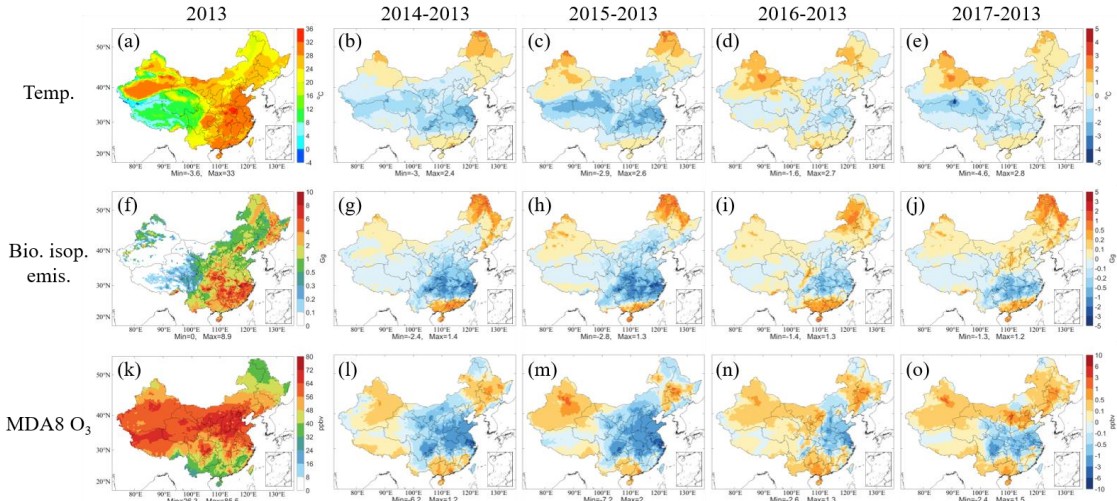

**Figure 5: The simulated daytime averaged temperature at a height of 2 m (Temp., the top row) and total biogenic isoprene emissions (Bio. isop. emis., the central row) in summer over China in 2013 from the base simulations, and their changes in 2014, 2015, 2016 and 2017 relative to 2013. The bottom row (MDA8 O₃) shows the simulated summer surface MDA8 O₃ concentrations in 2013 from the base simulation, and their changes due to variations in biogenic emissions in 2014, 2015, 2016 and 2017 relative to 2013.**




**Figure 6: The simulated averaged temperature (Temp.) and specific humidity (Humidity) at a height of 2 m, potential velocity at a**
**height of ~300 hPa (Wind/PV), planetary boundary layer (PBL) height, total clouds fraction (Clouds), and accumulated precipitation**
**(Precip.) in the daytime in summer of 2013 from the base simulation (the left column), and their changes in 2017 relative to 2013 (the**
**central column). The right column shows the changes in simulated summer surface MDA8 O$_3$ concentrations due to variations in**
**temperature, specific humidity, wind fields, PBL height, clouds and precipitation in 2017 relative to 2013. PVU is the potential**
**vorticity unit (1 PVU=10$^{-6}$ km$^2$ kg$^{-1}$ s$^{-1}$).**




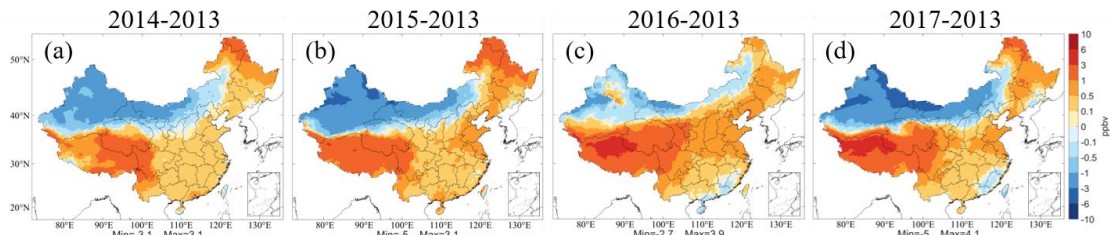

**Figure 7: Changes in the simulated summer surface MDA8 O₃ concentrations due to variations in long-range transport over China in 2014, 2015, 2016, and 2017 relative to 2013.**