# Peer review of "Worsening urban ozone pollution in China from 2013 to 2017 - Part"

_Atmospheric Chemistry and Physics, 2019_

## Referee Comment (RC1) · Anonymous Referee #1 · 3 Feb 2020

This paper provides a thorough analysis of the impact of meteorological variability on observed ozone changes across China from 2013 to 2017. The analysis is sound, for the most part, but there are a few inaccuracies that need to be addressed, as described below. Once these items are addressed I think the paper would be acceptable for publication in ACP.

Major comments: 1) The panels in Figure 1 are entirely too small and need to be increased by at least a factor of two, and rearranged on the page so that they fit. I had to enlarge the images on my computer to 400% and even then they were difficult to understand as the resolution was poor. Each panel has an inset in the lower right

corner, which doesn't seem to provide any information. These insets are distracting and should be removed. Likewise, the panels in Figures 3, 5 and 7 are also too small. For these figures you can expand the size of each panel by about 15% if you place the color bars underneath the panel, and move the labels on the left of the panels to positions above the panels. You can also delete the latitude and longitude labels, which aren't necessary. Then if you allow the panels to fill the full width of the page you should be able to make them significantly larger.

2) It would be helpful to place these 2013-2017 surface ozone changes in China within the context of broader trends across Asia, as well as long-term trends in the region of China. For example, Gaudel et al. use IAGOS observations to show that ozone in the lower and mid-troposphere has increased above China, India and Southeast Asia since 1994. Xu et al. show the long-term positive trend at Mt Waliguan, and Sun et al. show the positive trend at Mt. Tai. Wang et al. show the increase of ozone at Hok Tsui when transport is from the South China Sea. And Ziemke et al. show satellite retrievals that demonstrate a board increase of tropospheric column ozone across Asia and the tropics.

Gaudel, A, et al. 2018. Tropospheric Ozone Assessment Report: Present-day distribution and trends of tropospheric ozone relevant to climate and global atmospheric chemistry model evaluation. Elem Sci Anth, 6: 39. DOI: https://doi.org/10.1525/elementa.291

Sun, L, Xue, L, et al. 2016. Significant increase of summertime ozone at Mount Tai in Central Eastern China. Atmos. Chem. Phys. 16: 10637–10650. DOI: https://doi.org/10.5194/acp-16-10637-2016

Wang, T., Dai, J., Lam, K. S., Poon, C. N., and Brasseur, G. P. (2019), Twenty‐five years of lower tropospheric ozone observations in subtropical East Asia: The influence of emissions and weather patterns, Geophysical Research Letters, 46, https://doi.org/10.1029/2019GL084459

Xu, W, Lin, W, Xu, X, Tang, J, Huang, J, Wu, H and Zhang, X. 2016. Long-term trends of surface ozone and its influencing factors at the Mt Waliguan GAW station, China–Part 1: Overall trends and characteristics. Atmos. Chem. Phys. 16: 6191–6205. DOI: https://doi.org/10.5194/acp-16-6191-2016

Ziemke, J. R., Oman, L. D., Strode, S. A., Douglass, A. R., Olsen, M. A., McPeters, R. D., Bhartia, P. K., Froidevaux, L., Labow, G. J., Witte, J. C., Thompson, A. M., Haffner, D. P., Kramarova, N. A., Frith, S. M., Huang, L.-K., Jaross, G. R., Seftor, C. J., Deland, M. T., and Taylor, S. L.: Trends in global tropospheric ozone inferred from a composite record of TOMS/OMI/MLS/OMPS satellite measurements and the MERRA-2 GMI simulation , Atmos. Chem. Phys., 19, 3257-3269, https://doi.org/10.5194/acp-19-3257-2019, 2019.

3) Section 3.5 What is meant by "potential velocity"? Do you mean potential vorticity? Potential vorticity has long been used as in indicator of stratospheric intrusions into the upper and mid-troposphere, where it works very well, but it just doesn't work for the lower troposphere or the surface because the signal decays by the time the intrusion reaches the lower troposphere (if it ever reaches the lower troposphere). Linking an increase of ozone at the surface to an increase of PV in the upper troposphere is just speculation. How do you know the ozone reaching the surface is from the stratosphere? Couldn't it just be ozone from the mid-troposphere? (as shown by the IAGOS profiles in Gaudel et al. 2018, there is plenty of ozone in the mid-troposphere above China during the summer months) To provide a convincing argument that there was an increase of stratospheric ozone reaching the surface you will have to implement a conserved stratospheric ozone tracer in both MOZART and in CMAQ to see if there really is an increase of this tracer at the surface (see the papers by Meiyun Lin at NOAA GFDL, or papers by Andreas Stohl using the FLEXPART model). If you can't run a tracer all you can say is that there is likely an increase of ozone transport from the free troposphere to the surface, but you don't have any way of knowing if the ozone is from the mid-troposphere or if it's from the stratosphere.

[Figure]

4) Line 304 It would be helpful to treat humidity in a consistent manner throughout the paper. In Table 1 you report values of relative humidity, while in Figure 5 your show specific humidity. Why show both types of humidity? From an ozone chemistry perspective specific humidity is most important because it scales with water vapor concentration. Relative humidity isn't useful for understanding ozone photochemistry due to its non-linear relationship to water vapor concentration.

5) Line 322 This claim that precipitation can remove ozone is incorrect. The modeling study by Meleux et al. vaguely implies that precipitation removes ozone, but they don't give any mechanism or explanation, and this claim goes against the long established fact that ozone has very low solubility in water (Wesely et al., 1981). I can't think of any experimental studies that have shown that rain removes ozone from the air, although some studies have shown that chemicals in water (such as the ocean) can react with ozone if air bubbles are mixed into the ocean, or lakes (see the review by Monks et al., 2015, Atmos. Chem. Phys., 15, 8889–8973, 2015, www.atmos-chem-phys.net/15/8889/2015/ doi:10.5194/acp-15-8889-2015)

Wesely, M. L., Cook, D. R., and Williams, R. M.: Field measurement of small ozone fluxes to snow, wet bare soil, and lake water, Bound.-Lay. Meteorol., 20, 459–471, doi:10.1007/bf00122295, 1981.

Minor comments:

Line 45 Well, it's not the relative humidity value that is important, but rather the number of water vapor molecules that are available. It would be best to replace relative humidity with water vapor.

Line 48 Change "Cloud has" to "Clouds have"

Line 49 I'm not sure what you mean by "cleaning efficiency". Please use another term.

Line 50 How does the wet removal process increase ozone? Ozone is not water soluble. Is something else being removed by precipitation, which would otherwise destroy

ozone?

Line 65 If you are going to report ozone values in units of ppb, rather than in micrograms per cubic meter, you cannot use the term "concentration". Instead, please use mixing ratio.

Line 81 implications (plural) observational data

Line 82 . . .based on the observations.

Line 104 Would sound better as: The equations for these statistical parameters can be found in Fan et al. (2013).

Line 114 . . .which is a few grids cells smaller. . .

Line 157-158 Please see how I modified the following sentence to improve the English. The authors can make similar changes throughout the document. Original: "Like the temperature, the simulated relative humidity was also slightly under-predicted and had a high correlation coefficient with the observation." Corrected: "Like temperature, the simulated relative humidity values were also slightly under-predicted and had a high correlation coefficient with the observations."

Line 164 . . . conditions on ozone levels.

Line 205 . . .emissions on ozone changes. . .

Line 217 . . .could be comparable to or. . .

Line 312 I don't understand what is being said here: "Possible reasons for the ozone increase with the increase in the PBL height include low primary pollutant concentrations with the development of PBL" Are you saying that fresh emissions of NO can destroy ozone close to the surface in an urban environment, but if you have deep vertical mixing you can spread the NO vertically, which then limits ozone destruction at the surface?

Line 317 What is cleaning efficiency?

Line 365 Change "decreased" to "decrease"

Lines 383-386 These last two sentences should be revised. The first sentence is rather long and cumbersome and can be shortened as shown below. The second sentence is making a recommendation to policy-makers (by using the word "should") and does not belong in a scientific paper. However, it's perfectly fine to state how your results might be useful to policy-makers, by replacing "should" with "could" as shown below. "It is therefore necessary to consider meteorological variability when assessing the effectiveness of emission control policies on changes in the levels of ozone (and other air pollutants) in different cities and/or regions of China. Such an approach could be useful for the development of future air pollution mitigation policies."

Figure 2, caption The second sentence is difficult to understand. The following change would help: In panel (b) only environmental monitoring sites (493) with data available in all 5 years are presented.

---

## Referee Comment (RC2) · Anonymous Referee #2 · 13 Mar 2020

This manuscript presented a comprehensive modeling analysis on the surface ozone trends over China during 2013-2017. Significant ozone increases have been observed in China over this period in spite of the strong emission control actions implemented. Better understanding the drivers of these trends is of great scientific importance. The authors have conducted an ensemble of numerical simulations using the WRF-CMAQ air quality model to interpret these surface ozone trends, in particular, quantifying the role of meteorology in this manuscript. The results showed that the model had some success in reproducing the Chinese ozone increase trends, supporting the use of it to assess contributions from changes in anthropogenic emissions vs. changes in meteorology. The results further emphasized the importance of interannual variations

in meteorology affecting the recent surface ozone trends in China.

This is an important study, representing a great step to understand the drivers of interannual changes in summertime surface ozone pollution in China. The manuscript is well organized and written, and the methodology and results sound solid. I recommend publish after the following comments been addressed.

**Specific comments:**

1) Page 5, Line 120-125:

Some recent studies have suggested that the ozone increases in China since 2013 were largely driven by the concurrent decreases in PM2.5 levels and the resulting changes in heterogenous HO2 uptake by aerosol surfaces (Li et al., 2019a, 2019b). Since the model applied in this study reproduced the observed ozone increases, did the results support the important role of heterogenous reactions? Although the authors may discuss this issue in the second paper, I suggest put some sentences in this paper in the context of these recent findings.

Reference:

Li, K., et al., Anthropogenic drivers of 2013-2017 trends in summer surface ozone in China, P Natl Acad Sci USA, 116, 422-427, 2019.

Li, K. et al., A two-pollutant strategy for improving ozone and particulate air quality in China, Nature Geoscience, 12, 906-910, 2019.

2) Page 8, Figure 4:

The year 2013 seems to be a special year with particularly low ozone values, for example, as can be seen from Figure 4a) over Beijing (the BTH region). If the 2013 data point was removed from the linear trend calculation, then no trend was observed for Beijing. This is also the case for Guangzhou (Figure 4c) and the long-range transport ozone influences (Figure 7). Can you comment on this?

3) Page 10, first paragraph of section 3.5:
Figure 6 showed that the 2013-2017 changes in wind significantly increased surface ozone over most regions of China, and the authors attributed the ozone increases to enhanced transport from the lower stratosphere. This is not clear to me. It may explain some of the surface ozone increases in the northern and central China as argued by the PV changes, but how about the southern China where I think stratospheric ozone influences would be low at surface? It is not clear that enhanced ozone transport from the lower stratosphere could lead to 6-10 ppbv surface ozone increases in the southern China. I wonder whether changes in horizontal winds still contribute there, e.g., changes in the wind speed and the summer Asian monsoon. Please clarify.

4) Page 10, Line 301-302:
The statement "we found that the impact of temperature via the change in the chemical reaction rate was more significant than that via the change in biogenic emissions from 2013 to 2017" need to be more quantitative. It is difficult to read from Figure 5 and Figure 6 (the color bars are too small). It might depend on regions. I suggest compare their values averaged over the key regions and over China.

5) Page 12, Line 342-345:
The statements here seem to imply that transport of PAN led to the long-range transport of ozone influences. How about transport of ozone itself? Which one is the main pathway? One way to quantify and to separate the influences is to conduct a simulation fixing PAN in the 2013 chemical boundary conditions, yet I do not want to push the authors to do more model simulations. Can you explain the issue with present analyses and results?

---

## Author Comment (AC1) · 14 Apr 2020

This paper provides a thorough analysis of the impact of meteorological variability on observed ozone changes across China from 2013 to 2017. The analysis is sound, for the most part, but there are a few inaccuracies that need to be addressed, as described below. Once these items are addressed I think the paper would be acceptable for publication in ACP.

Response: We thank the referee for providing a thoughtful and detailed review of our paper. The referee's comments have helped to improve this manuscript. Below, we provide a point-by-point response to the referee's comments and summarize the changes that have been made in the revised manuscript.

Major comments:

[Comment]: 1. The panels in Figure 1 are entirely too small and need to be increased by at least a factor of two, and rearranged on the page so that they fit. I had to enlarge the images on my computer to 400% and even then they were difficult to understand as the resolution was poor. Each panel has an inset in the lower right corner, which doesn't seem to provide any information. These insets are distracting and should be removed. Likewise, the panels in Figures 3, 5 and 7 are also too small. For these figures you can expand the size of each panel by about 15% if you place the color bars underneath the panel, and move the labels on the left of the panels to positions above the panels. You can also delete the latitude and longitude labels, which aren't necessary. Then if you allow the panels to fill the full width of the page you should be able to make them significantly larger.

Response: Thanks for this comment. Regarding Figure 1, we have redrawn each panel with larger circles indicating the observational values. The composition of Figure 1 has been rearranged to enlarge the images to make them clearer. Following the referee's suggestion, we have removed the inset in each panel and placed the color bar underneath the panel in Figures 1, 3, 5, and 7. The labels on the left in Figures 3 and 5 have been changed from horizontal to vertical to save the horizontal space. We keep the latitude and longitude labels because we think this geographic information is important for readers to distinguish China's midlatitude areas (25°N to 40°N), southern (south of 25°N) and northern China (north of 40°N), which are mentioned in our manuscript. Similar changes in enlarging the panels have been made in Figures 2 and 6. Lastly, we have provided all figures with a higher resolution in the revised manuscript.

[Comment]: 2. It would be helpful to place these 2013-2017 surface ozone changes in China within the context of broader trends across Asia, as well as long-term trends in the region of China. For example, Gaudel et al. use IAGOS observations to show that ozone in the lower and mid-troposphere has increased above China, India and Southeast Asia since 1994. Xu et al. show the long-term positive trend at Mt Waliguan, and Sun et al. show the positive trend at Mt. Tai. Wang et al. show the increase of ozone at Hok Tsui when transport is from the South China Sea. And Ziemke et al. show satellite retrievals that demonstrate a board increase of tropospheric column ozone across Asia and the tropics.

Gaudel, A, et al. 2018. Tropospheric Ozone Assessment Report: Present- day distribution and trends of tropospheric ozone relevant to climate and global atmospheric chemistry model evaluation. Elem Sci Anth, 6: 39. DOI: https://doi.org/10.1525/elementa.291

Sun, L, Xue, L, et al. 2016. Significant increase of summertime ozone at Mount Tai in Central Eastern China. Atmos. Chem. Phys. 16: 10637–10650. DOI: https://doi.org/10.5194/acp-16-10637-2016

Wang, T., Dai, J., Lam, K. S., Poon, C. N., and Brasseur, G. P. (2019), Twen- tyâ Řˇfive years of lower tropospheric ozone observations in subtropical East Asia: The influence of emissions and weather patterns, Geophysical Research Letters, 46, https://doi.org/10.1029/2019GL084459

Xu, W, Lin, W, Xu, X, Tang, J, Huang, J, Wu, H and Zhang, X. 2016. Long-term trends of surface ozone and its influencing factors at the Mt Waliguan GAW station, China– Part 1: Overall trends and characteristics. Atmos. Chem. Phys. 16: 6191–6205. DOI: https://doi.org/10.5194/acp-16-6191-2016

Ziemke, J. R., Oman, L. D., Strode, S. A., Douglass, A. R., Olsen, M. A., McPeters, R. D., Bhartia, P. K., Froidevaux, L., Labow, G. J., Witte, J. C., Thompson, A. M., Haffner, D. P., Kramarova, N. A., Frith, S. M., Huang, L.-K., Jaross, G. R., Seftor, C. J., Deland, M. T., and Taylor, S. L.: Trends in global tropospheric ozone inferred from a composite record of TOMS/OMI/MLS/OMPS satellite measurements and the MERRA- 2 GMI simulation, Atmos. Chem. Phys., 19, 3257-3269, https://doi.org/10.5194/acp- 19-3257-2019, 2019.

Response: Our study intended to look into the effect of meteorology and emission changes on the recent surface ozone increase in China. We thank the referee for suggesting the above papers on troposphere ozone. We have made some changes in the first paragraph of the Introduction section in the revised manuscript.

Revision in the main text:

1) Line 30-33:

"With rapid urbanization and economic development, the ozone concentrations in the troposphere have increased in the past decades over most regions of Asia, including China (Gaudel et al., 2018; Sun et al., 2016; Wang et al., 2019c; Xu et al., 2016; Ziemke et al., 2019), and ground-level ozone pollution has become a major concern in China's urban and industrial regions (Wang et al., 2017; Verstraeten et al., 2015)."

[Comment]: 3. Section 3.5 What is meant by "potential velocity"? Do you mean potential vorticity? Potential vorticity has long been used as in indicator of stratospheric intrusions into the upper and mid-troposphere, where it works very well, but it just doesn't work for the lower troposphere or the surface because the signal decays by the time the intrusion reaches the lower troposphere (if it ever reaches the lower troposphere). Linking an increase of ozone at the surface to an increase of PV in the upper troposphere is just speculation. How do you know the ozone reaching the surface is from the stratosphere? Couldn't it just be ozone from the mid-troposphere? (as shown by the IAGOS profiles in Gaudel et al. 2018, there is plenty of ozone in the mid-troposphere above China during the summer months) To provide a convincing argument that there was an increase of stratospheric ozone reaching the surface you will have to implement a conserved stratospheric ozone tracer in both MOZART and in CMAQ to see if there really is an increase of this tracer at the surface (see the papers by Meiyun Lin at NOAA GFDL, or papers by Andreas Stohl using the FLEXPART model). If you can't run a tracer all you can say is that there is likely an increase of ozone transport from the free troposphere to the surface, but you don't have any way of knowing if the ozone is from the mid-troposphere or if it's from the stratosphere.

Response: Thanks for this valuable comment. What we mean is potential vorticity (PV) rather than "potential velocity". We have corrected this problem throughout the manuscript. We agree with the referee that it is unsuitable to link the ozone increase at the surface in the low-lying regions of eastern China to an increase in PV in the upper troposphere directly. However, for the Qinghai-Tibetan Plateau of western China with terrain heights > 3 km, PV analysis works well as demonstrated by many previous studies. Based on the PV results, we think that transport from the upper troposphere can explain in part the wind-induced increase in surface ozone in the western highlands. For eastern China (including southern region), we notice that the wind speeds decreased from 2013 to 2017, which would help the accumulation of ozone and ozone precursors and therefore increase ozone concentrations. Referee #2 also suggested that the ozone increases due to wind field changes could be attributed to the changes in horizontal transport. By examining the wind directions in the two years, we did not find evidence for change in vertical transport from the free troposphere to the surface or change in horizontal transport within the model domain. We have revised this section and clarified the possible reasons for the increasing ozone concentrations in terms of the changes in wind

speed, horizontal transport, and vertical transport. In Figure 6, we replaced the panels about PV with those on wind speed.

Revision in the main text:

1) Line 22-23:

   "The results show that the wind field change made a significant contribution to the increase in surface ozone over many parts of China."

2) Line 291-299:

   "Notable increases in MDA8 $O_3$ in western and eastern China due to the change in wind fields were identified, which contributed significantly to the meteorology-induced increasing ozone (Fig. 3h). In the Qinghai-Tibetan Plateau of western China whose terrain heights are greater than 3 km, the significant increase in the MDA8 $O_3$ mixing ratio (3 to 9 ppbv) due to wind change from 2013 to 2017 can be attributed in part to the enhanced downward transport from the upper troposphere as indicated by the increase in the potential vorticity (PV) (Fig. S6). In eastern China, the increase in $O_3$ level can be explained by the decrease in the wind speeds (Fig. 6h), which helps the accumulation of $O_3$ and its precursors and then increases ozone concentrations. There is no strong evidence for the change in the vertical transport from the free troposphere to the surface in eastern China and the horizontal transport from other regions within the modeling domain between these two years, according to the wind data (Fig. S7)."

3) Line 360-361:

   "The results show that the changes in the wind fields made a significant contribution to the increase in surface ozone levels over many parts of China."

4) Line 382-384:

   "The increase in MDA8 $O_3$ in Qinghai-Tibetan Plateau from 2013 to 2017 was ascribed to enhanced downward transport from the upper troposphere."

5) Line 646-647 (the caption of Figure 6):

   "wind speed at a height of 10 m (Wind)"

[Comment]: 4. Line 304 It would be helpful to treat humidity in a consistent manner throughout the paper. In Table 1 you report values of relative humidity, while in Figure 5 your show specific humidity. Why show both types of humidity? From an ozone chemistry perspective specific humidity is most important because it scales with water vapor concentration. Relative humidity isn't useful for understanding ozone photochemistry due to its non-linear relationship to water vapor concentration.

Response: We agree that the specific humidity is more useful for understanding ozone formation than the relative humidity. However, in regular weather observing networks, ambient humidity is generally measured and reported as relative humidity rather than specific humidity. Therefore, we compared the simulated relative humidity with the observed values to evaluate the modeling results of humidity. When investigating the impact of humidity on ozone, we changed to specific humidity because it is simulated and reported by the meteorological model. We have clarified this point in the Methods section of the revised manuscript.

Revision in the main text:

1) Line 150-152:

   "Here we used specific humidity rather than relative humidity because the specific humidity, which scaled with water vapor concentrations and was simulated by the model, was more useful for understanding the ozone formation chemistry."

[Comment]: 5. Line 322 This claim that precipitation can remove ozone is incorrect. The modeling study by Meleux et al. vaguely implies that precipitation removes ozone, but they don't give any mechanism or explanation, and this claim goes against the long established fact that ozone has very low solubility in water (Wesely et al., 1981). I can't think of any experimental studies that have shown that rain removes ozone from the air, although some studies have shown that chemicals in water (such as the ocean) can react with ozone if air bubbles are mixed into the ocean, or lakes (see the review by Monks et al., 2015, Atmos. Chem. Phys., 15, 8889–8973, 2015, www.atmos-chem-phys.net/15/8889/2015/ doi:10.5194/acp-15-8889-2015)

Wesely, M. L., Cook, D. R., and Williams, R. M.: Field measurement of small ozone fluxes to snow, wet bare soil, and lake water, Bound.-Lay. Meteorol., 20, 459–471, doi:10.1007/bf00122295, 1981.

Response: We agree that ozone is not water-soluble, and the amount of it removed by precipitation is limited. However, ozone precursors, such as $NO_2$, have relatively high solubility in water and can be removed by precipitation (Seinfeld and Pandis, 2006), which then decreases the ozone concentration (Shan et al., 2008). We have corrected this claim to "Although precipitation can decrease ozone concentrations via the scavenging of ozone precursors (Seinfeld and Pandis, 2006; Shan et al., 2008) …" in Line 326-327 in the revised manuscript.

Reference:

Seinfeld, J. H., and Pandis, S. N.: Atmospheric Chemistry and Physics-from Air Pollution to Climate Change, John Wiley & Sons, New Jersey, 2006.

Shan, W., Yin, Y., Zhang, J., and Ding, Y.: Observational study of surface ozone at an urban site in East China, Atmos Res, 89, 252-261, https://doi.org/10.1016/j.atmosres.2008.02.014, 2008.

Minor comments:

[Comment]: 6. Line 45 Well, it's not the relative humidity value that is important, but rather the number of water vapor molecules that are available. It would be best to replace relative humidity with water vapor.

Response: Thanks for this valuable suggestion. We agree with the referee that it is the water vapor that is important, rather than the relative humidity value. We have replaced "relative humidity" with "water vapor" in Line 46 in the revised manuscript.

[Comment]: 7. Line 48 Change "Cloud has" to "Clouds have"

Response: Thanks for this suggestion. We have changed "Cloud has" into "Clouds have" in Line 49 in the revised manuscript.

[Comment]: 8. Line 49 I'm not sure what you mean by "cleaning efficiency". Please use another term.

Response: Thanks for this comment. We have replaced "cleaning efficiency" with "scavenging of oxidants". This sentence has been changed into "Clouds have also been shown to decrease ozone concentrations via aqueous-phase chemistry and photochemistry, which enhances scavenging of oxidants and reduces the oxidative capacity of the troposphere" in Line 49-50 in the revised manuscript.

[Comment]: 9. Line 50 How does the wet removal process increase ozone? Ozone is not water soluble. Is something

else being removed by precipitation, which would otherwise destroy ozone?

Response: This is related to our response to comment 5, the ozone precursors (e.g., $NO_2$) rather than ozone can be removed by precipitation, which then decreases the ozone concentration. We have corrected the original statement to "precipitation decreases the ozone concentration via the wet removal of ozone precursors (Seinfeld and Pandis, 2006; Shan et al., 2008)" in Line 51-52 in the revised manuscript.

[Comment]: 10. Line 65 If you are going to report ozone values in units of ppb, rather than in micrograms per cubic meter, you cannot use the term "concentration". Instead, please use mixing ratio.

Response: Thanks for the valuable comment. We have changed "concentration" to "mixing ratio" in Line 66 in the revised manuscript. We also carefully went through the document and corrected similar issues throughout the revised manuscript.

[Comment]: 11. Line 81 implications (plural) observational data Line 82 . . .based on the observations.

Response: Thanks for pointing out these typos. We have changed "implication" into "implications" in Line 82, changed "observation data " into "observational data" in Line 82, and changed "based on the observation data" into "based on the observations" in Line 84 in the revised manuscript.

[Comment]: 12. Line 104 Would sound better as: The equations for these statistical parameters can be found in Fan et al. (2013).

Response: Thanks for this suggestion. We have changed "The calculation equations of these statistical parameters can be found in Fan et al. (2013)" into "The equations for these statistical parameters can be found in Fan et al. (2013)" in Line 104-105 in the revised manuscript.

[Comment]: 13. Line 114 . . .which is a few grids cells smaller. . .

Response: Here, it means the CMAQ modeling domain is a few grids cells smaller. To make it clearer, we have changed this sentence "The CMAQ modeling domain covers all the land area of China and the surrounding regions, which is a few grids smaller than the WRF modeling domain to reduce the effect of the meteorological boundary from the WRF model" into "The CMAQ modeling domain, which is a few grids smaller than the WRF modeling domain to reduce the effect of the meteorological boundary from the WRF model, covers all the land areas of China and the surrounding regions" in Line 114-116 in the revised manuscript.

[Comment]: 14. Line 157-158 Please see how I modified the following sentence to improve the English. The authors can make similar changes throughout the document. Original: "Like the temperature, the simulated relative humidity was also slightly under-predicted and had a high correlation coefficient with the observation." Corrected: "Like temperature, the simulated relative humidity values were also slightly under-predicted and had a high correlation coefficient with the observations."

Response: Thanks for this valuable suggestion. We have changed this sentence into "Like temperature, the simulated relative humidity values were also slightly under-predicted and had a high correlation coefficient with the observations" in Line 161-162 in the revised manuscript. Also, we carefully went through this manuscript and made

similar changes.

[Comment]: 15. Line 164 . . . conditions on ozone levels. Line 205 . . .emissions on ozone changes. . . Line 217 . . .could be comparable to or. . .

Response: Thanks for these valuable suggestions. We have changed "the ozone level" to "ozone levels" in Line 168-169, changed "to ozone changes" to "on ozone changes" in Line 212, and changed "comparable with" into "comparable to" in Line 225 in the revised manuscript.

[Comment]: 16. Line 312 I don't understand what is being said here: "Possible reasons for the ozone increase with the increase in the PBL height include low primary pollutant concentrations with the development of PBL" Are you saying that fresh emissions of NO can destroy ozone close to the surface in an urban environment, but if you have deep vertical mixing you can spread the NO vertically, which then limits ozone destruction at the surface?

Response: Thanks for this valuable comment. That is what we want to express. To make it clearer, we have changed this statement in the revised manuscript.

Revision in the main text:

1) Line 315-318:

"Possible reasons for the ozone increase with the increase in the PBL height include lower NO concentration at the urban surface due to the deep vertical mixing, which then limits ozone destruction and increases ozone concentrations (He et al., 2017), and more downward transport of ozone from the free troposphere where ozone is higher than the near-surface (Sun et al., 2009)."

[Comment]: 17. Line 317 What is cleaning efficiency?

Response: Similar to our response to comment 8, we have replaced "cleaning efficiency" with "scavenging of oxidants" in Line 321 in the revised manuscript.

[Comment]: 18. Line 365 Change "decreased" to "decrease"

Response: Thanks for pointing out this typo. We have changed "decreased" into "decrease" in Line 369 in the revised manuscript.

[Comment]: 19. Lines 383-386 These last two sentences should be revised. The first sentence is rather long and cumbersome and can be shortened as shown below. The second sentence is making a recommendation to policy-makers (by using the word "should") and does not belong in a scientific paper. However, it's perfectly fine to state how your results might be useful to policy-makers, by replacing "should" with "could" as shown below. "It is therefore necessary to consider meteorological variability when assessing the effectiveness of emission control policies on changes in the levels of ozone (and other air pollutants) in different cities and/or regions of China. Such an approach could be useful for the development of future air pollution mitigation policies."

Response: Thanks for these valuable suggestions. We have replaced these last two sentences with "It is therefore necessary to consider meteorological variability when assessing the effectiveness of emission control policies on changes in the levels of ozone (and other air pollutants) in different cities and/or regions of China. Such an approach

could be useful for the development of future air pollution mitigation policies" in Line 387-389 in the revised manuscript.

[Comment]: 20. Figure 2, caption The second sentence is difficult to understand. The following change would help: In panel (b) only environmental monitoring sites (493) with data available in all 5 years are presented.

Response: Thanks for this valuable suggestion. We have changed the sentence "As for the observation, only environmental monitoring sites (493) with data available in all 5 years are presented" to "In panel (b), only environmental monitoring sites (493) with data available in all 5 years are presented" in Line 623 in the revised manuscript.

---

## Author Comment (AC2) · 14 Apr 2020

This manuscript presented a comprehensive modeling analysis on the surface ozone trends over China during 2013-2017. Significant ozone increases have been observed in China over this period in spite of the strong emission control actions implemented. Better understanding the drivers of these trends is of great scientific importance. The authors have conducted an ensemble of numerical simulations using the WRF-CMAQ air quality model to interpret these surface ozone trends, in particular, quantifying the role of meteorology in this manuscript. The results showed that the model had some success in reproducing the Chinese ozone increase trends, supporting the use of it to assess contributions from changes in anthropogenic emissions vs. changes in meteorology. The results further emphasized the importance of interannual variations in meteorology affecting the recent surface ozone trends in China.

This is an important study, representing a great step to understand the drivers of interannual changes in summertime surface ozone pollution in China. The manuscript is well organized and written, and the methodology and results sound solid. I recommend publish after the following comments been addressed.

Response: We thank the referee for providing a thoughtful review of our paper and the recognition of our work. The referee's comments have helped to improve this manuscript. Below, we provide a point-by-point response to the referee's comments and summarize the changes that have been made in the revised manuscript.

Specific comments:

[Comment]: 1. Page 5, Line 120-125:

Some recent studies have suggested that the ozone increases in China since 2013 were largely driven by the concurrent decreases in PM2.5 levels and the resulting changes in heterogenous $HO_2$ uptake by aerosol surfaces (Li et al., 2019a, 2019b). Since the model applied in this study reproduced the observed ozone increases, did the results support the important role of heterogenous reactions? Although the authors may discuss this issue in the second paper, I suggest put some sentences in this paper in the context of these recent findings.

Reference:

Li, K., et al., Anthropogenic drivers of 2013-2017 trends in summer surface ozone in China, P Natl Acad Sci USA, 116, 422-427, 2019.

Li, K. et al., A two-pollutant strategy for improving ozone and particulate air quality in China, Nature Geoscience, 12, 906-910, 2019.

Response: Thanks for pointing out this and suggesting the two recent papers. We have discussed the impact of heterogeneous reactions on the ozone changes in the second paper (Liu and Wang, 2020), which also supported the significant role of heterogeneous reactions in the increasing urban ozone concentration across China. Following the referee's suggestion, we have revised the manuscript and placed the incorporation of comprehensive heterogeneous chemistry into the CMAQ model within the context of these recent findings.

Revision in the main text:

1) Line 121-124:

"The original CMAQ model includes the heterogeneous reactions of only $NO_2$, $NO_3$, and $N_2O_5$ on aerosol surfaces. Recent studies (Li et al., 2019a; Li et al., 2019b) have suggested that the heterogeneous reactions on aerosol surfaces, mainly the uptake of $HO_2$, played a significant role in the increasing $O_3$ concentrations in China from 2013 to 2017. To better simulate the effects of aerosol on ozone via heterogeneous reactions, …"

Reference:

Liu, Y., and Wang, T.: Worsening urban ozone pollution in China from 2013 to 2017 – Part 2: The effects of emission changes and implications for multi-pollutant control, Atmos. Chem. Phys. Discuss., 2020, 1-27, 10.5194/acp-

2020-53, 2020.

[Comment]: 2. Page 8, Figure 4:

The year 2013 seems to be a special year with particularly low ozone values, for example, as can be seen from Figure 4a) over Beijing (the BTH region). If the 2013 data point was removed from the linear trend calculation, then no trend was observed for Beijing. This is also the case for Guangzhou (Figure 4c) and the long-range transport ozone influences (Figure 7). Can you comment on this?

Response: Thanks for this comment. For some regions, such as Beijing and Guangzhou, the year 2013 does seem to be a special year with particularly low ozone concentration. But this is not the case in other regions such as Shanghai and Chengdu, in which the ozone concentrations in 2013 were higher than those in 2014. Figure 3a-d also shows that the changes in MDA8 $O_3$ during 2014-2017 relative to 2013 were increases or decreases, depending on regions and years. Such characteristics are attributed to the complex and varying roles of meteorology in ozone changes, which are highlighted in this paper.

As for the impact of long-range transport on ozone changes, we think the year 2013 is also not a special year with the lowest influence from the long-range ozone transport compared with other years. We remove the 2013 data and plot Figure R1 below to show the changes in the simulated MDA8 $O_3$ due to variations in long-range transport in 2015, 2016, and 2017 relative to 2014. Similar increases in ozone are still found, which supports the increasing contribution from long-range transport to ozone in these years. We have added this statement to the revised manuscript.

Revision in the main text:

1) Line 231-232:

"As shown in Fig. 4, the changes in observed MDA8 $O_3$ varied in cities and years, which were generally captured by the model"

2) Line 346-348:

"Increases in ozone levels were also found if we compared the changes in MDA8 $O_3$ due to variations in chemical boundary conditions relative to 2014."

[Figure]

Figure R1 Changes in the simulated summer surface MDA8 $O_3$ mixing ratios due to variations in long-range transport over China in 2015, 2016, and 2017 relative to 2014.

[Comment]: 3. Page 10, first paragraph of section 3.5:

Figure 6 showed that the 2013-2017 changes in wind significantly increased surface ozone over most regions of China, and the authors attributed the ozone increases to enhanced transport from the lower stratosphere. This is not clear to me. It may explain some of the surface ozone increases in the northern and central China as argued by the PV changes, but how about the southern China where I think stratospheric ozone influences would be low at surface? It is not clear that enhanced ozone transport from the lower stratosphere could lead to 6-10 ppbv surface ozone increases in the southern China. I wonder whether changes in horizontal winds still contribute there, e.g., changes in the wind speed and the summer Asian monsoon. Please clarify.

Response: Thanks for this comment. Referee #1 raised a similar concern. We think that the PV analysis works well for Qinghai-Tibetan Plateau of western China with terrain heights > 3km. But for other regions (eastern and southern), surface ozone increases are due to a decrease in wind speeds, which would help the accumulation of ozone and ozone precursors in these regions and increase ozone concentrations. By examining the wind directions in the two years, we did not find evidence for change in vertical transport from the free troposphere to the surface or change in horizontal transport within the model domain. We have revised this section and clarified the possible reasons for the increasing ozone concentrations in terms of the changes in wind speed, horizontal transport, and vertical transport. In Figure 6, we replaced the panels about PV with those about wind speed.

Revision in the main text:

1) Line 22-23:

   "The results show that the wind field change made a significant contribution to the increase in surface ozone over many parts of China."

2) Line 291-299:

   "Notable increases in MDA8 $O_3$ in western and eastern China due to the change in wind fields were identified, which contributed significantly to the meteorology-induced increasing ozone (Fig. 3h). In the Qinghai-Tibetan Plateau of western China whose terrain heights are greater than 3 km, the significant increase in the MDA8 $O_3$ mixing ratio (3 to 9 ppbv) due to wind change from 2013 to 2017 can be attributed in part to the enhanced downward transport from the upper troposphere as indicated by the increase in the potential vorticity (PV) (Fig. S6). In eastern China, the increase in $O_3$ level can be explained by the decrease in the wind speeds (Fig. 6h), which helps the accumulation of $O_3$ and its precursors and then increases ozone concentrations. There is no strong evidence for the change in the vertical transport from the free troposphere to the surface in eastern China and the horizontal transport from other regions within the modeling domain between these two years, according to the wind data (Fig. S7)."

3) Line 360-361:

   "The results show that the changes in the wind fields made a significant contribution to the increase in surface ozone levels over many parts of China."

4) Line 382-384:

   "The increase in MDA8 $O_3$ in Qinghai-Tibetan Plateau from 2013 to 2017 was ascribed to enhanced downward transport from the upper troposphere."

5) Line 646-647 (the caption of Figure 6):

   "wind speed at a height of 10 m (Wind)"

[Comment]: 4. Page 10, Line 301-302:

The statement "we found that the impact of temperature via the change in the chemical reaction rate was more

significant than that via the change in biogenic emissions from 2013 to 2017" need to be more quantitative. It is difficult to read from Figure 5 and Figure 6 (the color bars are too small). It might depend on regions. I suggest compare their values averaged over the key regions and over China.

Response: Thanks for this suggestion. In the revised manuscript, we have enlarged the panels in Figures 5 and 6 and provided high-resolution figures, which can help to compare the impacts of changes in temperature and biogenic emissions on MDA8 $O_3$. Figure 5o and 6c are put together in Figure R2 below. We also compared their values averaged in four typical cities, namely Beijing, Shanghai, Guangzhou, and Chengdu, in Figure R3 (also being provided in the supplementary material as Figure S8). We noted that the impact of temperature and biogenic emissions on ozone could be different in some regions (e.g., opposite impacts in Beijing), which can be explained by the transport of emitted pollutants. However, the increases (decreases) in MDA8 $O_3$ due to changes in temperature were generally higher than those due to changes in biogenic emissions, which supported the statement in our manuscript. Revision in the main text:

1) Line 304-306:

"However, comparing Fig. 5o to Fig. 6c, we found that the impact of temperature via the change in the chemical reaction rates was generally more significant than that via the change in biogenic emissions from 2013 to 2017 (also see Fig. S8 for the quantitative comparisons in different cities)."

[Figure]

Figure R2 Changes in the simulated summer surface MDA8 $O_3$ mixing ratios due to the changes in (a) temperature and (b) biogenic emissions in 2017 relative to 2013.

[Figure]

Figure R3 (Figure S8) Changes in the simulated summer surface MDA8 O₃ mixing ratios due to the changes in temperature and biogenic emissions in 2017 relative to 2013 in Beijing, Shanghai, Guangzhou, and Chengdu.

[Comment]: 5. Page 12, Line 342-345:

The statements here seem to imply that transport of PAN led to the long-range transport of ozone influences. How about transport of ozone itself? Which one is the main pathway? One way to quantify and to separate the influences is to conduct a simulation fixing PAN in the 2013 chemical boundary conditions, yet I do not want to push the authors to do more model simulations. Can you explain the issue with present analyses and results?

Response: Thanks for pointing out this issue. The original statement does seem to indicate that the transport of PAN is the only pathway affecting the long-range transport of ozone, which needs to be corrected. Previous observation and modeling studies (West et al., 2009; Wild et al., 2004) have suggested that ozone and its precursors, namely NO$_x$ (or its carrier PAN), VOCs and CO, can be transported a long distance and then affect the ozone concentration in remote regions. We have corrected this statement in the revised manuscript.

We also think that investigating the pathway of long-range O₃ transport is an interesting and important topic and agree that one way to address this issue is to conduct many simulations fixing different pollutants individually in the chemical boundary conditions and compare the simulation results with each other. As the present study focuses on the impact of meteorological parameters, we prefer to address which chemical contributed most during long-range transport in a future study.

Revision in the main text:

1) Line 347-348:

"Ozone and its precursors can be transported a long distance and then affect surface O₃ in remote regions (West et al., 2009; Wild et al., 2004)."

Reference:

West, J. J., Naik, V., Horowitz, L. W., and Fiore, A. M.: Effect of regional precursor emission controls on long-range ozone transport - Part 1: Short-term changes in ozone air quality, Atmos Chem Phys, 9, 6077-6093, 10.5194/acp-9-6077-2009, 2009.

Wild, O., Pochanart, P., and Akimoto, H.: Trans-Eurasian transport of ozone and its precursors, 109, 10.1029/2003jd004501, 2004.